# Effect of Intense Pulsed-Light Treatment Using a Novel Dual-Band Filter in Patients with Meibomian Gland Dysfunction

**DOI:** 10.3390/jcm11133607

**Published:** 2022-06-22

**Authors:** Mincheol Kim, Jisang Min

**Affiliations:** Department of Ophthalmology, Kim’s Eye Hospital, College of Medicine, Konyang University, Seoul 35365, Korea; mconly77@kimeye.com

**Keywords:** dry eye, intense pulsed-light therapy, vascular filter, meibomian gland dysfunction

## Abstract

Background: This study evaluates the effect of intense pulsed-light (IPL) treatment in patients with meibomian gland dysfunction (MGD) using a novel dual-band filter (vascular filter, 530–650 nm and 900–1200 nm) and compares it with the effect and discomfort during treatment using a conventional filter. Methods: The medical records of 89 patients (89 eyes) with MGD who underwent IPL treatment were reviewed. Patients treated with the vascular filter or conventional 590 nm filter were designated as Group A or Group B, respectively. Patients underwent IPL treatment four times every four weeks. Ocular surface disease index (OSDI) scores, dry eye (DE), and MGD parameters were determined before the first IPL treatment and after the fourth IPL treatment. Visual analog scale (VAS) scores were obtained at every IPL treatment. OSDI, DE and MGD parameters, and VAS were compared between the groups. Results: OSDI, DE, and MGD parameters improved after the four IPL treatments in both groups. There were no significant differences, between the groups, in OSDI, DE, and MGD parameters, before the first IPL treatment and after the fourth IPL treatment. VAS at each of the IPL treatments was lower in Group B than in Group A. Conclusion: IPL treatment using the novel vascular filter for patients with MGD is effective compared with conventional IPL treatment for MGD patients.

## 1. Introduction

Meibomian gland dysfunction (MGD) is a disease that occurs in up to 70% of the population, particularly in Asia [1]. MGD can affect meibomian gland secretions in terms of quality and/or quantity, possibly resulting in an unstable tear film [2]. Consequently, it can cause symptoms such as dryness, eye irritation, foreign body sensation, burning, watering, and fatigue [3]. Conventional treatments for MGD are warm compresses, lid massage, anti-inflammatory ointments, and artificial tears [4]. Despite the variety of treatment options available, many patients with MGD are refractory and unsatisfied with the treatment.

Intense pulsed-light (IPL) treatment has been applied for hypertrichosis, cavernous hemangiomas, venous malformations, telangiectasia, port-wine stains, and other pigmented lesions [5]. Toyos et al. [6] first introduced IPL treatment in the field of ophthalmology, and patients with facial rosacea had significant improvements in dry eye (DE) symptoms after IPL treatment. Many previous studies demonstrated that IPL treatment is effective for the improvement of both subjective symptoms and objective findings in patients with mild-to-moderate MGD or DE [7,8,9,10].

The mechanisms underlying IPL treatment in patients with MGD include superficial blood vessel destruction, meibum fluidification, epithelial turnover downregulation, photomodulation, and antimicrobial effect [10]. Among these mechanisms, the main one involved in IPL treatment for MGD patients is considered to be superficial blood vessel destruction, and IPL therapy is known to be the only treatment that can improve eyelid signs, including superficial blood vessel formation [11,12].

The M22 Optima device (Lumenis, Yokneam, Israel) is one of the most widely used IPL machines for MGD treatment [6,13,14,15,16,17,18,19,20]. The M22 has various filters and can be used according to the treatment purpose, and a novel dual-band filter (vascular filter, wavelengths 530–650 nm and 900–1200 nm), designed particularly for fine telangiectasia treatment, was introduced recently. However, even though the main mechanism of IPL treatment for MGD patients is superficial blood vessel destruction, there are no studies on this new filter to date, and no study has compared the treatment effects between the conventional filter and the new dual-band filter.

Therefore, in this study, we investigated the effect of IPL treatment using the novel dual-band filter and compared it with the effect of IPL treatment with the conventional filter used in MGD patients.

## 2. Materials and Methods

### 2.1. Patients

This study was conducted with the approval of the institutional review board (No. KEH 2021-11-016-002). We analyzed the medical records of consecutive patients diagnosed with MGD between March 2021 and December 2021 who received a series of four IPL treatments. MGD patients were diagnosed according to previously described criteria [21,22]: (1) at least one symptom among eye fatigue, discharge, foreign body sensation, dryness, stickiness sensation, pain, epiphora, itching, redness, heaviness, glare, excessive blinking, burning, or ocular discomfort on awakening; (2) at least one abnormal eyelid margin, such as vascular congestion, anterior or posterior replacement of mucosal skin junctions, and irregular eyelid margins; and (3) plugged meibomian gland orifices and poorly expressible meibum in the target eye. IPL treatment was performed on patients who were refractory to or unsatisfied with conventional treatments such as artificial tears, warm compresses, eyelid rubs, or topical/systemic antibiotics. Patients who met the following inclusion criteria were enrolled: (1) age of more than 18 years and (2) completion of four consecutive IPL treatments at 4-week intervals. Patients who met the following exclusion criteria were excluded: (1) missing DE and meibomian gland examination results before the first IPL treatment or after the fourth IPL treatment; (2) systemic diseases that can influence DE disease; (3) oral or topical retinoid use; (4) intraocular surgery in the past 6 months; (5) botulinum toxin or filler injection in the past month; (6) uncontrolled ocular disease; or (7) dark skin type, such as Fitzpatrick skin type V or VI [23].

### 2.2. IPL Procedure

Before IPL treatment, patients were asked to clean up their faces to remove makeup. After a sufficient amount of ultrasound gel was applied to the target skin area, the clinician placed the Jaeger lid plate (Katena Products, Denville, NJ, USA) within the conjunctival sac to protect the eye. In the case of IPL treatment, an M22 Optima device (Lumenis, Yokneam, Israel) was used, and the duration and interval were set as 6.0 msec and 60.0 msec, respectively; additionally, a 6 mm cylindrical light guide was applied [20]. In the case of the filter, a novel dual-band filter (vascular filter) or a 590 nm filter was used. The fluence was set according to Fitzpatrick skin types (13–19 J/cm^2^), as described in previous studies [14,16,20]. Subsequently, 12 IPL pulses were then applied to the upper and lower eyelids [16,20]. Once the IPL treatment was finished, meibomian gland expression was performed with an Arita Meibomian Gland Compressor (Katena Products, Denville, NJ, USA).

### 2.3. Clinical Assessment

In all patients, DE and MGD parameters and the ocular surface disease index (OSDI) scores were obtained before the first and after the fourth IPL treatments. In addition, the pain and discomfort that occurred during the IPL treatment were measured using visual analog scale (VAS) scores, and they were determined at the first (IPL#1), second (IPL#2), third (IPL#3), and fourth (IPL#4) IPL treatments (Figure 1).

The DE parameters were as follows: type I Schirmer test (ST), tear break-up time (TBUT) test, and corneal staining scores (CFS). MGD parameters were as follows: lid margin abnormality score (LAS), meibomian gland examinations such as meibum expressibility (ME), meibum quality (MQ), and lipid layer thicknesses (LLT). A standard paper strip (Eagle Vision, Memphis, TN, USA) was placed for 5 min without topical anesthesia on the third of the mid-lateral portion of the lower fornix, and the length of the wetting column was determined as ST. A single fluorescein strip (Haag-Streit International, Koniz, Switzerland) was placed over the inferior tear meniscus with a drop of preservative-free normal saline to obtain CFS and TBUT. TBUT was measured after several times of blinking; the measurements were repeated three times, and the average TBUT was calculated. CFS was obtained according to the corneal staining pattern of the Oxford Schema [24]. The LipiView interferometer (TearScience, Morrisville, NC, USA) was used to measure LLT. The lid margins and meibomian glands were examined and measured under a slit-lamp microscope after other measurements were performed. The LAS value was either 0 (absent) or 1 (present) for lid margin irregularity, vessel engorgement, plugged meibomian glands, and anterior or posterior mucocutaneous junction displacement [25]. After the application of digital pressure over five lower meibomian glands, ME was measured as the number of expressible glands: grade 0, all 5 glands expressible; grade 1, 3–4 glands expressible; grade 2, 1–2 glands expressible; and grade 3, none of the glands expressible [25]. MQ was also measured based on the following scores: 0, clear; grade 1, cloudy; grade 2, cloudy with granular debris; and grade 3, toothpaste-like. The scores of each of the 8 glands were summed to determine a total score (maximum score, 24) [25].

Patients were divided into two groups based on the filter type used in the IPL treatments. Patients who underwent IPL treatment with the vascular filter were designated in Group A, and patients who received IPL treatment with the conventional 590 nm filter were designated in Group B.

The DE and MGD parameters of the right eye and OSDIs of Groups A and B were obtained. Changes in the obtained DE and MGD parameters and OSDI before IPL#1 and after IPL#4 were compared. Furthermore, the DE and MGD parameters and OSDI of Groups A and B were compared before IPL#1 and after IPL#4. Additionally, the VAS scores of IPL#1, IPL#2, IPL#3, and IPL#4 were compared between Groups A and B.

In the case of vessel engorgement among LAS, an additional analysis was performed. The changes in vessel engorgement before IPL#1 and after IPL#4 were compared. Additionally, vessel engorgement of Groups A and B were compared before IPL#1 and after IPL#4.

The two groups were compared using the independent *t*-test for continuous variables and the chi-square test for categorical variables. SPSS, version 25.0 (IBM Corp., Armonk, NY, USA) was used for all statistical analyses. *p*-values of <0.05 were considered statistically significant.

## 3. Results

### 3.1. Demographics

Table 1 shows the demographic characteristics of patients in Groups A and B. Group A consisted of 47 patients (47 eyes, 19 males; average age: 56.52 ± 14.95 years), and Group B consisted of 44 patients (44 eyes, 14 males; average age: 53.23 ± 11.58 years). There were no significant differences between the two groups in the demographics.

### 3.2. Changes in DE and MGD Parameters and OSDI before IPL#1 and after IPL#4 in Each Group

Table 2 shows the changes in DE and MGD parameters and OSDIs of Groups A and B before IPL#1 and after IPL#4. There was no significant change in ST and LLT of both groups before IPL#1 and after IPL#4. TBUT, CFS, LAS, ME, MQ, and OSDI had improved after IPL#4 in both groups.

### 3.3. Comparison of OSDI and DE and MGD Parameters between Groups A and B before IPL#1 and after IPL#4

Table 3 shows the DE and MGD parameters and OSDI of Groups A and B before IPL#1 and after IPL#4. There were no significant differences before IPL#1 in OSDI, DE, and MGD parameters between Groups A and B. Additionally, there were no differences after IPL#4 in OSDI, DE, and MGD parameters between Groups A and B.

### 3.4. Comparison of Pain and Discomfort between Groups A and B during Each IPL Treatment

Table 4 shows the VAS scores at IPL#1, IPL#2, IPL#3, and IPL#4 for Groups A and B. The VAS scores at IPL#1, IPL#2, IPL#3, and IPL#4 in Group A were significantly higher than those in Group B. Additionally, the average VAS score was significantly higher in Group A than in Group B. There were no patients who had serious side effects after IPL treatments.

### 3.5. Intragroup and Intergroup Comparison of Vessel Engorgement Findings

Table 5 shows vessel engorgement findings of Groups A and B before IPL#1 and after IPL#4. In both groups, the number of patients without vessel engorgement of the lid was significantly increased. There were no differences in the ratio of the number of patients with vessel engorgement of the lid and without vessel engorgement of the lid.

## 4. Discussion

In this study, patients who received IPL treatment using a vascular filter experienced an improvement in DE and MGD parameters, as well as OSDI. Additionally, there were no significant differences in values obtained before IPL#1 and after IPL#4 between patients treated with a new vascular filter and those treated with a conventional filter. Treatment using the novel vascular filter improved the signs and symptoms of MGD patients, and the treatment effect was comparable to that of the conventional 590 nm filter. The pain and discomfort occurring during the IPL treatment using the novel vascular filter were greater than those occurring with the conventional 590 nm filter.

There are two types of novel dual-band filters according to the wavelengths: 530–650 nm and 900–1200 nm (vascular filter) and 400–600 nm and 800–1200 nm (acne filter). There are studies that have successfully treated facial acne vulgaris safely with an acne filter [26,27]. Many studies report the successful treatment of MGD patients using the M22 device [6,13,14,15,18,19,20,28,29]. However, all these studies used the existing 590 nm filter, and studies reporting the use of the novel dual-band filters in the treatment of IPL in MGD patients are rare. In the ophthalmology fields, only one study reported IPL treatment using an acne filter in MGD patients [30]. However, to the best of our knowledge, there are no studies on IPL treatment using a vascular filter in the ophthalmology field, and this study is the first to report the evaluation of the effect of IPL treatment using a vascular filter in MGD patients. As in the previous study on acne filter utility [30], IPL treatment using the novel dual-band filter emitting 530–650 nm and 900–1200 nm is effective to treat MGD patients.

Hemoglobin is known to have absorption spectra with double peak absorption around 400 nm and 550–600 nm [31]. Because the acne filter includes both peak wavelengths, the effect of the acne filter is likely to be superior to that of other filters, such as vascular filters. However, there has been no study comparing the therapeutic effects of IPL treatment using an acne filter and IPL treatment using other filters such as a vascular filter in MGD patients. Therefore, it is necessary to compare the therapeutic effects of IPL treatment using a vascular filter and that using an acne filter in MGD patients. In patients with MGD, IPL treatment is a relatively new treatment option. Therefore, additional detailed studies on the efficacy of this treatment method in a particular mode, specific filters, and titration of light energy are required.

The VAS score of patients treated with IPL treatment using a vascular filter was higher than that with a conventional 590 nm filter. As with hemoglobin, which is the treatment target for patients with MGD, melanin is a chromophore found in the epidermis, and light absorption by melanin can induce pain [32]. The novel vascular filter contains a shorter notched wavelength section than the conventional 590 nm filter; hence, it seems that it is because the light absorption by melanin on the skin surface is higher. Although the VAS of the novel vascular filter is higher than that of the conventional filter, the average VAS of IPL#1, #2, #3, and #4 is very low (approximately 1.0); therefore, it can be considered that the pain during IPL treatment using the novel vascular filter is tolerable. In the M22 Optima device, which was used in this study, there is an advanced optimal pulse technology (AOPT) mode, and this mode divides each pulse into several sub-pulses of low fluence and emits the group of sub-pulses into the treatment area. In the case of pain, the application of the AOPT mode is expected to further reduce pain during IPL treatment. However, there are few studies about pain during IPL treatment at the present time [32,33]. Recently, Thaysen-Petersen et al. [33] conducted a study on hair removal using IPL and reported that the higher the fluorescence or dark skin, the higher the pain during IPL treatment. However, there have been no studies on pain during IPL treatment using this filter at present and the present study is the first on pain during IPL treatment to the best of the authors’ knowledge. In addition, it is thought that additional research on pain during IPL treatment according to modulation, such as filtering, during IPL treatment is needed in the future.

Previously, many MGD treatments have been introduced, of which only IPL improved the superficial vessel ablation of the eyelid. There are studies reporting successful outcomes of facial telangiectasia with IPL treatment [33,34,35,36]. In these studies, filters specific to wavelengths of 500–550 nm [33], 500–600 nm [34], or the vascular [36] filter were used, and double pulse [33,36] or single pulse [34] was set. The lid abnormality score of the patients who received IPL treatment using both a vascular filter and a 590 nm filter improved. In addition, the vessel engorgement of the lid had decreased after IPL treatment when using both a vascular filter and a 590 nm filter. The vascular filter is a filter specialized for the treatment of vascular lesions of the skin and has a wavelength in the notched range of 530–650 nm. Therefore, it was expected that the vessel engorgement of the lid would decrease more when treated with the vascular filter than when treated with the 590 nm filter. However, there was no significant difference in the degree of vessel engorgement of the lid between the two groups before IPL treatment and after IPL treatment. As in other studies reporting treatment of MGD with IPL [6,14,15,18,19,25,28,29], IPL was set in the triple pulse mode. In studies reporting treatment of facial telangiectasia with IPL, the treatment target vessel is located on the skin surface. However, in this study, the target of IPL treatment is the meibomian gland, which is in the tarsal plate, located deeper into the skin surface. Therefore, IPL treatment for MGD patients is usually set to triple pulse for deep penetration. This setting of IPL for deep penetration might have no difference in vessel engorgement change between patients treated with IPL using a vascular filter and those treated with IPL using a 590 nm filter.

This study has certain limitations, which should be considered. First, it had a retrospective study design. Second, the follow-up period was limited to 4 weeks after the final treatment. Longer follow-up periods are needed to evaluate long-term changes in a patient’s eyelid skin temperature. Third, randomized controlled clinical trials or well-designed cohort studies are required to confirm the treatment effect of patients with MGD after IPL treatment using a vascular filter. Fourth, several factors, such as differences in skin melanin levels and local skin inflammatory mediators for each patient that could influence the study results were not considered in this study. Therefore, a study using the two types of filters in the same patient will require a conventional filter on one eye and a novel dual filter on the other eye.

## 5. Conclusions

In conclusion, IPL treatment using a novel dual-band filter yielded significant improvement in DE and MGD signs and symptoms as well as in DE symptoms; moreover, its treatment effect was comparable to that of IPL treatment using a conventional filter.

## Figures and Tables

**Figure 1 jcm-11-03607-f001:**
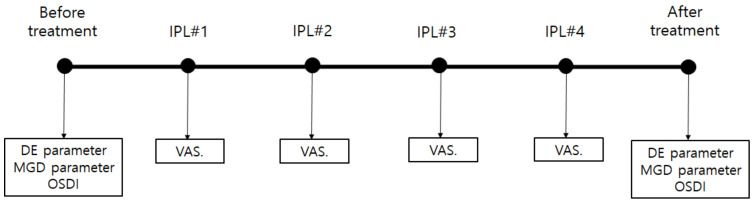
Brief schedule of IPL treatment and clinical evaluations. DE and MGD parameters and the OSDI are obtained before the first IPL treatment and after the fourth IPL treatment. Abbreviations: IPL, intense pulsed light; DE, dry eye; IPL#1, first IPL treatment; IPL#2, second IPL treatment; IPL#3, third IPL treatment; IPL#4, fourth IPL treatment; MGD, Meibomian gland dysfunction; OSDI, ocular surface disease index; VAS, visual analog scale.

**Table 1 jcm-11-03607-t001:** Demographics of the patients in each group.

	Group A	Group B	*p*-Value
Patients (n)	47 (47 eyes)	44 (44 eyes)	
Age (years)	56.52 ± 14.95	53.23 ± 11.58	0.251
Sex (M:F)	19:28	14:30	0.393

Data are shown as mean ± standard deviation. Statistical analyses were performed using the chi-square test for categorical variables and independent *t*-test for continuous variables. Abbreviations: M, male; F, female.

**Table 2 jcm-11-03607-t002:** Results of changes in DE and MGD parameters obtained before and after intense pulsed-light treatment in Groups A and B.

	Group A			Group B		
	Before IPL#1	After IPL#4	*p*-Value	Before IPL#1	After IPL#4	*p*-Value
ST	11.45 ± 8.05	11.85 ± 9.04	0.694	14.36 ± 11.73	16.67 ± 10.50	0.694
TBUT	3.25 ± 1.14	5.45 ± 2.43	<0.001	3.25 ± 1.14	5.45 ± 2.43	<0.001
CFS	2.09 ± 4.29	0.36 ± 0.71	<0.001	1.14 ± 0.83	0.67 ± 0.86	<0.001
LLT	72.48 ± 27.18	71.85 ± 23.57	0.264	72.48 ± 27.18	71.85 ± 23.57	0.264
LAS	2.04 ± 0.56	0.82 ± 0.64	<0.001	1.89 ± 0.66	0.86 ± 0.74	<0.001
ME	1.56 ± 0.78	0.69 ± 0.69	<0.001	1.74 ± 0.94	0.74 ± 0.78	<0.001
MQ	19.00 ± 6.55	9.93 ± 4.89	<0.001	19.00 ± 5.39	10.40 ± 6.11	<0.001
OSDI	33.71 ± 18.91	10.85 ± 10.62	<0.001	42.31 ± 18.03	20.21 ± 48.01	0.002

Data are shown as mean ± standard deviation. Statistical analysis was performed using independent *t*-tests. Abbreviations: IPL#1, first intense pulsed-light treatment; IPL#4, fourth intense pulsed-light treatment; ST, type 1 Schirmer test; TBUT, tear break-up time; CFS, corneal staining scores; LLT, lipid layer thickness; LAS, lid margin abnormality score; ME, meibum expressibility; MQ, meibum quality, OSDI, ocular surface disease index; DE, dry eye; MGD, meibomian gland dysfunction

**Table 3 jcm-11-03607-t003:** Comparison of DE and MGD parameters and OSDI obtained before IPL#1 and after IPL#4 in Groups A and B.

	Before IPL#1			After IPL#4		
	Group A	Group B	*p*-Value	Group A	Group B	*p*-Value
ST	11.45 ± 8.05	14.36 ± 11.73	0.174	11.85 ± 9.04	16.67 ± 10.50	0.072
TBUT	3.25 ± 1.14	3.25 ± 1.14	0.685	5.45 ± 2.43	5.45 ± 2.43	0.767
CFS	2.09 ± 4.29	1.14 ± 0.83	0.143	0.36 ± 0.71	0.67 ± 0.86	0.072
LLT	72.48 ± 27.18	72.48 ± 27.18	0.995	71.85 ± 23.57	71.85 ± 23.57	0.296
LAS	2.04 ± 0.56	1.89 ± 0.66	0.247	0.82 ± 0.64	0.86 ± 0.74	0.815
ME	1.56 ± 0.78	1.74 ± 0.94	0.337	0.69 ± 0.69	0.74 ± 0.78	0.758
MQ	19.00 ± 6.55	19.00 ± 5.39	0.103	9.93 ± 4.89	10.40 ± 6.11	0.694
OSDI	33.71 ± 18.91	42.31 ± 18.03	0.190	10.85 ± 10.62	20.21 ± 48.01	0.200

Data are shown as mean ± standard deviation. Statistical analysis was performed using paired *t*-tests. Abbreviations: IPL#1, first intense pulsed-light treatment; IPL#4, fourth intense pulsed-light treatment; ST, type 1 Schirmer test; TBUT, tear break-up time; CFS, corneal staining scores; LLT, lipid layer thicknesses; LAS, lid margin abnormality score; ME, meibum expressibility; MQ, meibum quality, OSDI, ocular surface disease index; DE, dry eye; MGD, meibomian gland dysfunction.

**Table 4 jcm-11-03607-t004:** Visual analog scale scores for each intense pulsed-light treatment.

	Group A	Group B	*p*-Value
IPL#1	0.97 ± 0.90	0.38 ± 0.82	0.001
IPL#2	1.00 ± 0.96	0.29 ± 0.65	<0.001
IPL#3	1.15 ± 1.29	0.40 ± 0.71	<0.001
IPL#4	0.86 ± 1.15	0.19 ± 0.40	<0.001
Average of IPL#1, #2, #3, #4	1.00 ± 1.08	0.31 ± 0.66	<0.001

Data are shown as mean ± standard deviation. Statistical analysis was performed using independent *t*-tests. Abbreviations: IPL#1, first intense pulsed-light treatment; IPL#2, second intense pulsed-light treatment; IPL#3, third intense pulsed-light treatment; IPL#4, fourth intense pulsed-light treatment.

**Table 5 jcm-11-03607-t005:** Vessel engorgement findings of the lid before IPL#1 and after IPL#4 in Groups A and B.

	Before IPL#1	After IPL#4	*p*-Value
Group A (V: noV)	45:2	30:17	0.001
% of V	95.74	63.82	
Group B (V: noV)	42:2	26:18	<0.001
% of V	95.45	59.09	
*p*-value	0.999	0.381	

Statistical analyses were performed using the chi-square test for categorical variables and independent *t*-test for continuous variables. Abbreviations: IPL#1, first intense pulsed-light treatment; IPL#4, fourth intense pulsed-light treatment; v, number of patients with vessel engorgement of the lid; noV, number of patients without vessel engorgement of the lid; %, percentage.

## Data Availability

Not applicable.

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
