# Peer review of "Effect of Intense Pulsed-Light Treatment Using a Novel Dual-Band Filter in Patients with Meibomian Gland Dysfunction"

_jcm, 2022, doi:10.3390/jcm11133607_

Round 1

Reviewer 1 Report

reviewer compliments authors to under take critical analysis of Novel  dual  band filter (530-560 nm and 900 1200 nm)with conventional filter. Intense pulse light therapy has emerged as a promising tool to manage cases of MGD and OSD. Being  relatively newer mode of treatment, there is a definite need to undertake studies to carryout detailed evaluation of efficacy of equipment ,filters as well as modulation and titration  of light energy used in this particular mode of treatment. The present analysis fills gape in the existing literature .

Author Response

We fully agree with the reviewers' comments, and thank you for your comments on the paper. As commented by the reviewers, further detailed studies of this new treatment using Novel dual band filters are needed. Therefore, these contents have been added to the Discussion section (Line 253-256).

Reviewer 2 Report

Comparison of two different IPL (intense pulsed- light) treatments which differed in that an additional wavelength of light was used in the first or novel type (Dual filter).  There was no difference between clinical effects of the IPL types except for the VAS (Visual Analog Scale). VAS should be better explained  and a more extensive discussion provided as to why it may have decreased with the additional wavelength of light present in the novel treatment  (lines 254-266). Overall this was an well done retrospective study regarding IPL treatment of a common eye disease.  Ophthalmologists and dermatologists will find this of interest.  

Author Response

Thank you for your interest and opinion for this study. As described in the method (Table 4) and discussion (line 257-270) section, VAS of group A (patients treated IPL with vascular filter) was higher than group B (patients treated IPL with conventional 590 nm filter), however, the VAS of group A was relatively low (approximately 1.0 in average), and considered to be tolerable. Melanin is one of the chromophore mainly located in the epidermis, we believe that light absorption of the melanin will occur pain during IPL treatment. However, there are few studies about the pain occurring during IPL treatment. Recently, Thaysen-Petersen et al. conducted a study on hair removal using IPL, and reported that the higher the fluorescence or dark skin, the higher the pain during IPL treatment. However, there have been no studies on pain during IPL treatment according to the filter yet, and this study is the first study on pain during IPL treatment according to the filter to the knowledge of the authors. In addition, it is thought that additional research on pain during IPL treatment according to modulation such as filter during IPL treatment is needed in the future., and we have mentioned this in the discussion section (Line 270-277).

Reviewer 3 Report

I appreciate the details of the OSD and MGD evaluation . Despite these tests , there may be known factors (e.g melanin levels in the skin) or unknown factors (e.g local inflammatory mediators) that may influence the results of the treatments studied. Perhaps a structuring of the study by using the two types of filters in the same patient; one eye standard filter and to congener the dual filter , would eliminate intersubjective differences in response to treatment in parallel with a better subjective assessment of each type of treatment.

Author Response

I fully agree with the reviewers' comments, and thank you for your comments on the paper.

As the recommendation of the reviewer, the further study by using the two types of filters in the same patient will be needed. In addition, we are actively considering conducting these studies in the future. We have mentioned above in the limitation of discussion section (line 305-309).
